# Experimental and Numerical Study of the Mixed Lubrication Considering Boundary Film Strength

**DOI:** 10.3390/ma16031035

**Published:** 2023-01-24

**Authors:** Shengwei Zhang, Zhijun Yan, Ze Liu, Yuanyuan Jiang, Haocheng Sun, Shibo Wu

**Affiliations:** Marine Engineering College, Dalian Maritime University, Dalian 116026, China

**Keywords:** boundary film strength, mixed lubrication, CFD

## Abstract

For the influence of boundary film on the lubrication state of sliding friction pairs, a boundary film strength model was proposed that can comprehensively reflect the influences of film thickness, pressure, shear stress and temperature. The model parameters were obtained through fitting the test results. Then, a mixed lubrication model considering boundary film strength was established by coupling the boundary film strength model with the hydrodynamic lubrication model and the asperity contact model. The calculation program was developed using the Fortran language, which can effectively capture the tribological characteristics and action ratios of the fluid, boundary film and dry friction components. Simultaneously, the mixed lubrication model was applied to the journal bearing. A parametric analysis was performed to investigate the influences of different working conditions on lubrication performance. Under current operating conditions, the results show that: when the speed is above 200 r/min or the viscosity is higher than 0.09 Pa·s, the boundary film breakdown rate is almost 0 and the friction coefficient is lower than 0.02; when the roughness is reduced from 1.8 μm to 0.8 μm, the ultimate load of the journal bearing rises from 27 MPa to 36 MPa, an increase of about 33%; when the load exceeds 36 MPa or the temperature is higher than 100 °C, more than 25% of the boundary film breaks and the dry friction component accounts for more than 60% of the total friction, which leads to a sudden increase in the friction coefficient. Hence, the study of mixed lubrication considering boundary film strength provides theoretical guidance for accurately reflecting the actual lubrication state and improving the mechanical energy efficiency of friction pairs.

## 1. Introduction

When internal combustion engines, machine tools and other machinery are under heavy load or start/stop state, some of the friction pairs are usually in a mixed lubrication state [1]. In the mixed lubrication state, the fluid film and boundary film exist and act simultaneously between the interfaces, so the lubrication is the result of the combination of fluid lubrication and boundary lubrication [2,3,4,5]. At present, the research on the simulation methods of pure hydrodynamic lubrication and elastohydrodynamic lubrication (EHL) is relatively mature; however, the research on the simulation methods of boundary lubrication still needs to be improved [6,7]. The existence of boundary film can avoid metal-to-metal contact, thus ensuring that the friction pair is under a good lubrication state [8,9,10,11,12,13]. However, when the boundary film breaks due to mechanical or thermal effects, metal-to-metal direct contact will occur, which will sharply increase the friction coefficient and cause adhesive wear. Therefore, it is necessary to consider the influence of boundary film failure between surfaces on contact characteristics and lubrication characteristics when studying mixed lubrication, so as to reflect the lubrication state between interfaces more accurately.

The causes of boundary film breakdown are complex, and much exploration has been carried out. As early as 1939, Block [14] developed the flash temperature theory, which suggested that the breakdown of the boundary film was caused by the surface temperature reaching a critical value. In 1972, Czichos et al. [15,16] conducted a study on concentrated contact failure of lubrication, which showed that the critical damage loads of boundary films corresponding to different sliding velocities at a certain temperature were different. Then they proposed the concept of failure surface in 1974. In 1994, Kelly et al. [17] proposed a thermal model incorporating salient features of scuffing in a mixed lubrication state. In 2000, Wang et al. [18] investigated the boundary film strength from the perspective of distortion energy. They concluded that the bond between the lubricant molecules or atoms and the substrate would be destroyed not only by the thermal energy generated at high temperatures but also by the other energies, such as distortion energy generated by shear. In 2007, Wang et al. [19] proposed that the coverage of the boundary film was closely related to the heat generated by friction. In 2011, Ajayi et al. [20] proposed an adiabatic shear instability model to determine whether the failure of the boundary film occurs based on the rate of thermal softening due to the heat of plastic deformation and the rate of work hardening. In 2013, Li et al. [21] simulated the boundary film breakdown by coupling a thermal EHL model with a heat transfer model. In 2015, Wojciechowski et al. [22] presented a proposal for invariant precursors for boundary lubrication scuffing that considered the interaction between the rheological, morphological and physicochemical properties of contacting the surface’s layer. In 2016, the severe effect of shear stress on the boundary film was confirmed in a study by Zhang and Spikes [23]. In 2018, Xu et al. [24] proposed a boundary lubrication model considering the dynamic effect of boundary films by analyzing the generation rate and removal rate of boundary films. In 2019, Lee et al. [25] derived a boundary film failure model based on the principle of additive mass conservation and concluded that the depletion of the additive caused the failure of the boundary film during sliding. In 2021, Lyu et al. [26] suggested that the higher shear energy of the fluid would lead to the collapse of the boundary film based on the thermal fluid film thinning effect.

The above models are mainly based on boundary film failure’s occurrence and propagation mechanism. Because various friction systems have different mechanisms for the formation and failure of boundary films and it is difficult to determine the failure criteria through the test, the application of the above models is limited. Therefore, it is a feasible method to establish an empirical boundary film strength model based on the influences of the main parameters [27]. This paper developed an empirical boundary film strength model that considers film thickness, pressure, shear stress and temperature. Its parameters can be obtained from the test results, which is easier for engineering applications. This model combines the hydrodynamic lubrication model and the asperity contact model to establish a mixed lubrication model that considers boundary film strength. The mixed lubrication model is applied to the actual friction pair to investigate the tribological characteristics and action ratios of the fluid, boundary film and dry friction components under different working conditions. Additionally, the model is used to assess how speed, lubricant viscosity, surface roughness, load and temperature affect lubrication performance.

## 2. Test Method

### 2.1. Testing Machine

In this study, a friction testing machine has been designed to test the boundary film strength. Figure 1 and Figure 2 show the structure of the testing machine and the sample installation.

The upper sample was a stepped steel shaft, and the lower samples were two brass bars. The upper and lower sample surfaces were in vertical tangent contact, forming a two-point contact. The lower samples were installed in the oil box, and a floating support was connected under the oil box to ensure the two lower samples were in uniform contact with the upper sample. In the test, the friction force was measured by the sensor arranged at one side of the floating support and the normal force was measured by the sensor arranged at the bottom of the floating support. The normal force can be adjusted by the loading equipment. When the machine operated stably, oil films formed on the surfaces of the moving samples. By applying a constant voltage (5 V) between sample 1 and sample 2 and measuring the current in the circuit loop of lower sample 1, oil film, upper sample, oil film and lower sample 2, the change in oil film resistance can be measured in real-time to reflect the change of lubrication state.

### 2.2. Test Preparation

Table 1 shows the test parameters. The upper sample was a 45-steel (high-quality carbon structural steel with a carbon content of 0.45%) stepped shaft supported by bearing chocks. One side of the shaft was connected to the motor by a flexible coupling. The cylindrical surface in the middle of the stepped shaft was the working surface and had a diameter of 50 mm. It was polished with SiC sandpaper (2000 mesh) and alumina polishing solution. The two lower samples were 8 mm in diameter and made of H59 brass. The surface roughness of the samples was measured by a TR210 roughness meter, and CF4 10W-40 lubricant was used for the test.

### 2.3. Test Specifications

All the tests were based on the following specifications: (a) before each test the lower samples were ultrasonically cleaned for 15 min in ethanol; (b) the lower samples were installed and fixed in the oil box and an adequate amount of lubricant was added to the oil box to ensure the lubricant submerged the lower samples; (c) a heating resistor heated the lubricant; (d) the stable friction data for each test condition were measured after at least 10 min of running in; (e) after completing one test, the contact surface of the samples was disassembled and readjusted and the measurement was repeated three times to reduce randomness; (f) the oil box was cleaned, and the lubricant was replaced before the next test.

## 3. Boundary Film Strength Model

### 3.1. The Failure Features of the Boundary Lubrication

Since the contact resistance method can effectively characterize the lubrication state [28,29,30], the resistance signal was measured to ensure that the test can be performed in the boundary lubrication state. Figure 3 shows the lubrication state division based on the contact resistance method. As the load increases, the oil film becomes thinner. Meanwhile, the asperity contacts increase so the resistance decreases. When the contact resistance drops to a low level and is almost stable, the friction pair is considered to enter the boundary lubrication state.

For the test, load, speed and temperature were chosen as the primary variables to determine the features of boundary lubrication failure.

Figure 4 shows the variation of friction coefficient with load at different sliding speeds when the temperature is 55 °C. Corresponding to a specific sliding speed, there was an abrupt increase in the friction coefficient as the load increased. This reflects the lubrication state change. When the boundary film breaks, dry friction will occur, so the sudden increase in the friction coefficient is caused mainly by the boundary film breakdown. The figure also shows that the characteristic load related to the abrupt increase in the friction coefficient lowers as the sliding speed increases. When the sliding speed is 0.1 m/s, the boundary film breakdown occurs when loaded up to 170 N. When the speed is 1.0 m/s, the breakdown occurs when loaded only up to 80 N. This indicates that the load and sliding speed significantly affect the boundary film failure.

Figure 5 shows the variation of the friction coefficient with sliding speed under different loads at 55 °C. When the load is small (20, 40 and 60 N), the increase in sliding speed will not cause the boundary film to break. When the load is 100 N, an abrupt increase in the friction coefficient occurs when the sliding speed increases to 0.5 m/s. The reason for this phenomenon is that when the load is small, due to the oil film being thicker, even if the sliding speed is considerable, the boundary film is not easy to break. In contrast, when the load increases, the oil film becomes thinner and the shear stress increases with the sliding speed [31]. At this time, the boundary film is more susceptible to damage because of shear stress.

The tests measured the variation of the friction coefficient with load at five temperatures (35, 55, 80, 100 and 120 °C) and different speeds (0.1, 0.3, 0.5, 0.75 and 1.0 m/s). Figure 6 shows the variation of the friction coefficient with loads at different temperatures when the sliding speed is 0.3 m/s. It reflects the effect of temperature on boundary film failure. As the temperature increases, the characteristic load value corresponding to the sudden increase in friction coefficient decreases. When the temperature is 35 °C, the sudden increase in friction coefficient occurs when loaded up to 180 N. However, when the temperature increases to 120 °C, the sudden increase in friction coefficient occurs when loaded up to only 70 N. Therefore, besides the load and sliding speed, temperature also has a significant effect on the boundary film failure.

### 3.2. Model Fitting

For the working conditions corresponding to the boundary film failure in Section 3.1. Based on the Hamrock–Dowson film thickness equation [32] and the asperity contact model proposed by Greenwood and Tripp [33], the asperity contact pressure *Pa* and film thickness *h* are obtained when the boundary film fails. The oil film shear stress *τ* is modeled by the Erying model [34]. Figure 7 shows the relationship between the asperity contact pressure and oil film shear stress. It can be seen that the asperity contact pressure when the boundary film fails decreases with the increase in the oil film shear stress. In addition, the higher the temperature, the lower the asperity contact pressure with the same shear stress.

In summary of the test and calculation results, the boundary film failure is due to pressure, oil film shear and temperature. Therefore, the boundary film strength model can be expressed as a three-dimensional surface, as shown in Figure 8. When the working parameters are below the surface, the friction pair is in a better lubrication state; otherwise, the boundary film breaks down and the friction pair will enter a poor lubrication state. This surface can be expressed as a function of the asperity contact pressure *Pa* (MPa), the shear stress *τ* (MPa) of the oil film and the temperature *T* (°C):(1)$$P_{a}=S\left(T,\;\tau \right)=a\times T^{b}\times \tau^{c}+d$$

The statistics data from Figure 7 are fitted using MATLAB to get the model parameters, and the expression is as follows:(2)$$P_{a}=S\left(T,\;\tau \right)=362.1\times T^{-0.034}\times \tau^{-0.199}-496.9$$

The similarity coefficient is 0.9607, indicating a good agreement between this strength model and the test results.

## 4. Establishment of Mixed Lubrication Model

### 4.1. Basic Model

The boundary film strength model is applied to establish a mixed lubrication model. The Reynolds equation is used as the fundamental model for fluid lubrication. Assuming that the surface roughness follows the Gaussian distribution, the boundary film strength model is used as the criterion for the transition from boundary lubrication to dry friction.

Based on the established mixed lubrication model, a journal bearing is simulated and verified as the object. The diagram of the journal bearing structure is shown in Figure 9. R_1_ and R_2_ are the radii of the journal and bearing, m; c = R_2_ − R_1_ is the radius clearance, m; e is the eccentricity distance, m; *W* is the external load, N; U is the tangential linear velocity of journal surface, m/s; Ψ is the eccentricity angle; and *θ* is the circumferential coordinate starting from the bearing vertex.

For convenience, the lubrication interface is expanded along the circumference [35], x denotes the bearing circumferential coordinate, y denotes the bearing axial coordinate and z denotes the bearing radial coordinate. When cylindrical coordinates are used, set *x* = *R*_2_*θ* and select the dimensionless axial coordinate *Y* = *y*/*b*; dimensionless oil film pressure *P_h_* = *p_h_*/2UηR_2_; dimensionless oil film thickness *H* = *h*/c. The Reynolds equation can be written as:(3)$$\frac{\partial }{\partial \theta }\left(\varphi_{x}H^{3}\frac{\partial P_{h}}{\partial \theta }\right)+{\left(\frac{R}{b}\right)}^{2}\frac{\partial }{\partial Y}\left(\varphi_{y}H^{3}\frac{\partial P_{h}}{\partial Y}\right)=3\varphi_{c}\frac{\partial H}{\partial \theta }+3\sigma \frac{\partial \varphi_{s}}{\partial \theta }$$
where *φ_x_* and *φ_y_* are the pressure flow factors in the circumferential and axial directions, respectively, *φ_c_* is a dimensionless factor, *σ* is the integrated roughness of the two rough surfaces and *φ_s_* is the shear flow factor. The specific parameters and their meanings are described in Refs. [36,37].

The boundary conditions are as follows:(4)$${\left.P_{h}\right|}_{Y=+\frac{1}{2}}={\left.P_{h}\right|}_{Y=-\frac{1}{2}}=0$$
(5)$${\left.\frac{\partial P_{h}}{\partial \theta }\right|}_{\theta =\theta_{2}}=0,{\left.{\left.\;\;\;\;P_{h}\right|}_{\theta =0}=P_{h}\right|}_{\theta =\theta_{2}}=0$$
where *θ_2_* is the coordinate of oil film pressure boundary.

### 4.2. Asperity Contact

The load in the contact area under mixed lubrication is shared between the oil film pressure and the asperity contact pressure. In this study, the asperity contact model established by Greenwood and Tripp [33] is used. Define the film thickness ratio *λ* = 4(*λ* = *h*/*σ*) as the critical value for whether or not asperity contact occurs. The asperity contact pressure *Pa* and area of asperity contacts *Ac* are in Equations (6) and (7).
(6)$$P_{a}\left(h\right)=\left(\frac{16\sqrt{2}}{15}\right)\pi {\left(n\beta \sigma \right)}^{2}\cdot \sqrt{\frac{\sigma }{\beta }}\cdot E\cdot F_{2.5}\left(\lambda \right)$$
(7)$$A_{c}=\pi^{2}{\left(n\beta \sigma \right)}^{2}AF_{2.0}\left(\lambda \right)\;$$
where *A* is the nominal contact area, *n* is the asperity density and *β* is the asperity radius.

This study assumes that when the film thickness ratio *λ* ≥ 4, the asperities will not contact. When *λ* < 4 but the boundary film is not broken, the *μ_a_* (friction coefficient of the boundary film) is usually between 0.05 and 0.2 [6]. When the boundary film breaks, the friction coefficient will increase sharply to the dry friction level [38]. The friction coefficient (when the boundary film breaks) is defined as *f*_0_, and the value of *f*_0_ used in the paper is 0.5. Therefore, the friction coefficient between the asperities *f*_c_ can be expressed as the following piecewise function:(8)$$f_{c}=\left\{\begin{array}{c} 0\;\;\;\;\;\;\;\;\;\lambda \geq 4\\ \mu_{a\;\;}\;\;\;\;\;\lambda <4\;and\;P_{a}<S\left(T,\tau \right)\\ f_{0\;\;\;\;}\;\;\;\;\lambda <4\;and\;P_{a}\geq S\left(T,\tau \right) \end{array}\right.$$

The oil film and the asperities share the load so it is given by:(9)$$W=W_{h}+W_{a}\;$$
where *W_h_* denotes the load carried by the oil film and *W_a_* denotes the load carried by the asperities.

The load carried by the oil film *W_h_* in the x and y directions are *W_hx_* and *W_hy_*, respectively.
(10)$$\left\{\begin{array}{c} W_{hx}=\iint p_{h}\sin\left(\psi +\theta \right)dxdy\\ W_{hy}=\iint p_{h}\cos\left(\psi +\theta \right)dxdy \end{array}\right.$$
(11)$$W_{h}=\sqrt{W_{hx}{}^{2}+W_{hy}{}^{2}}$$

The load carried by the asperities *W_a_* in the x and y directions are *W_ax_* and *W_ay_*, respectively.
(12)$$\left\{\begin{array}{c} W_{ax}=\iint P_{a}\sin\left(\psi +\theta \right)dxdy\\ W_{ay}=\iint P_{a}\cos\left(\psi +\theta \right)dxdy \end{array}\right.$$
(13)$$W_{a}=\sqrt{W_{ax}{}^{2}+W_{ay}{}^{2}}$$

Friction also consists of two parts:(14)$$F=F_{f}+F_{c}$$
where *F_f_* is the fluid viscosity traction force, *F_c_* is the asperity contact friction. The calculations are shown in Equations (15) and (16):(15)$$F_{f}=\iint \tau_{f}\left(x,y\right)dxdy\;$$
(16)$$F_{c}=\iint f_{c}\times P_{a}\left(x,y\right)dxdy$$

Oil film shear stress is solved by the Eyring model:(17)τf=τ0arcsinhηγτ0 
where *τ*_0_ is the Eyring stress, Pa, *γ* is the shear rate, 1/s, *η* is lubricant dynamic viscosity, Pa·s.

Therefore, the friction coefficient can be calculated by Equation (18):(18)f=FW

### 4.3. Solution Process and Preliminary Verification

The solution process for the mixed lubrication model is illustrated in Figure 10. The program is based on the Fortran language.

To preliminary verify the mixed lubrication model, the calculation results in this paper are compared with those of Xu et al. [39]. The parameters are shown in Table 2, and Figure 11 shows the comparison results.

The comparison results show that the shape of the pressure distribution and the calculated pressure values at the center and b/4 are consistent, and the model established in this paper is preliminarily verified.

### 4.4. Experimental Verification

The tribological characteristics were tested by a testing machine for journal bearings to further verify the mixed lubrication model. Figure 12 shows its loading structure.

The friction data were measured at 55 °C and 70 °C. Table 3 gives the related parameters.

Figure 13 shows the comparison between the test results and the simulation results. By comparison, the measured and simulated friction coefficients show similar trends, confirming that the mixed lubrication model proposed in this paper can accurately reflect the tribological characteristics.

It should be noted that although the Greenwood and Tripp model is widely used to estimate mixing/boundary lubrication, recent studies suggest that this model may under-estimate the proportion of mixed/boundary friction in actual contact when the film thick-ness is relatively small [40,41,42]. Therefore, the accuracy of the model may decrease with a lower film thickness ratio.

## 5. Application of Mixed Lubrication Model

The mixed lubrication model established in this paper is applied to the journal bearing. The influences of different working conditions on lubrication performance are studied through systematic parameter analysis.

### 5.1. The Influence of Sliding Speed

Figure 14 reveals the effect of sliding speed on lubrication performance. The working conditions are as follows: load *W* = 36 MPa, temperature *T* = 55 °C, viscosity *η*_0_ = 0.05 Pa·s, roughness *σ* = 1.13 μm and speed *N* varies from 100 r/min to 300 r/min. From Figure 14a, it can be seen that the minimum oil film thickness *h_min_* and film thickness ratio increase as the speed increases and the *h_min_* rises from about 0.57 μm to 1.12 μm. As the oil film becomes thicker, the percentage of load carried by asperities decreases from 20% to 6.7% and the area of asperity contacts also decreases from 7.2% to about 2.2%, as shown in Figure 14b. Each friction component is shown in Figure 14c for the mixed lubrication state. With the increase in speed, the direct asperity contact and boundary film breakdown decrease. Meanwhile, the dry friction component decreases continuously and the total friction is gradually dominated by the fluid component and boundary film component, making the friction force decrease constantly. The friction coefficient in Figure 14d also shows the same change law. However, it is worth noting that when the speed rises from 100 r/min to 140 r/min, although the friction coefficient gradually decreases, the boundary film breakdown rate increases slowly. The reason for this phenomenon is that the shear stress in the oil film increases at this stage. According to the boundary film strength model, shear stress will lead to the boundary film breakdown.

From the above analysis, low speed is not conducive to the formation of oil film and cannot ensure a good lubrication state. An appropriate increase in speed can avoid the breakdown of the boundary film and the direct contact of asperities, reducing friction to improve mechanical efficiency and lubrication performance.

### 5.2. The Influence of Lubricant Viscosity

Figure 15 gives the effect of lubricant viscosity on lubrication performance. The working conditions are as follows: speed *N* = 100 r/min, load *W* = 36 MPa, temperature *T* = 55 °C, roughness *σ* = 1.13 μm and viscosity *η*_0_ varies from 0.03 to 0.12 Pa·s. From Figure 15a, it can be seen that the minimum oil film thickness *h_min_* and film thickness ratio increase together with the increase in viscosity. The *h_min_* increases from 0.42 μm to 0.98 μm, and the film thickness ratio increases from 0.37 to 0.87. As the viscosity increases, the percentage load carried by asperities decreases from 26.5% to about 9% and the area of asperity contacts decreases from 9.6% to about 3.1%, as shown in Figure 15b. The variation of the friction force and its components with viscosity can be seen in Figure 15c. When the viscosity is below 0.07, the friction force is 4500 N, mainly provided by the boundary film and dry friction. The dry friction component disappears as the viscosity increases, and the total friction decreases continuously. When the viscosity is 0.09, the friction force decreases to about 1800 N and is only provided by the fluid and boundary film, indicating that the friction pair is under a better lubrication state. According to Figure 15d, the coefficient of friction shows the same trend as friction force. When the viscosity is lower than 0.07, the breakdown rate of the boundary film is higher than 25% and the friction coefficient is more than 0.037. With the increase in viscosity, oil film with a higher bearing capacity is established, which reduces the contact of asperities and the breakdown of the boundary film. When the viscosity is higher than 0.08, the friction coefficient decreases significantly and the boundary film breakdown rate is almost 0. The friction coefficient reaches the lowest value of 0.012 when the viscosity is 0.12.

Therefore, under the current working conditions, improving viscosity can significantly improve the lubrication performance. It should be noted that, in practice, high viscosities can cause increased friction within the lubricant, leading to increased energy consumption and the oxidative deterioration of the lubricant.

### 5.3. The Influence of Roughness

The effect of surface roughness on lubrication performance is shown in Figure 16. The working conditions are as follows: speed *N* = 100 r/min, load *W* = 36 MPa, temperature *T* = 55 °C, viscosity *η*_0_ = 0.05 Pa·s and surface roughness *σ* varies from 0.2 μm to 2.0 μm. From Figure 16a, it can be seen that the minimum oil film thickness *h*_min_ increases as the surface becomes rougher. The reason for this phenomenon is that the increase in roughness makes the asperities carry a more significant portion of the load, which reduces the oil film pressure and thus increases the film thickness. It should be noted that although *h*_min_ increases, the film thickness ratio decreases from 1.85 for smooth surfaces to 0.4 for rough surfaces. At the same time, the percentage of load carried by asperities increases from nearly 0 to 35.5% and the area of asperity contacts increases from 1.2% to about 12.8%, as shown in Figure 16b. From Figure 16c, it can be visualized that the friction increases as the surface becomes rougher. The rising trend of friction is relatively flat when the roughness increases from 0.2 μm to 0.8 μm, and the boundary film and fluid mainly provide the friction. The friction increases rapidly from 1800 N to 12,400 N when the roughness exceeds 0.8 μm. The main component of friction, dry friction, reaches 10,000 N. The reason for this phenomenon is that as the surface becomes rougher, the contact pressure of the asperity increases. When the load and speed are fixed, the excessive contact pressure of the asperity will lead to the breakdown of the boundary film. As shown in Figure 16d, when the roughness is higher than 0.8 μm, the breakdown rate of boundary film increases from nearly 0 to 51% and the friction coefficient increases rapidly from 0.015 to 0.1. The variation of friction coefficient with load for different surface roughnesses is given in Figure 16e. When the surface roughness is 1.8 μm, the friction coefficient increases abruptly after 27 MPa, whereas when the roughness is 0.8 μm, the friction coefficient increases abruptly after the load reaches 36 MPa. 

If the ultimate load is defined as the load at which the friction coefficient abruptly increases, the ultimate load decreases as the roughness rises.

### 5.4. The Influence of Load

Figure 17 illustrates how load affects lubrication performance. The working conditions are as follows: speed *N* = 100 r/min, temperature *T* = 55 °C, viscosity *η*_0_ = 0.05 Pa·s, roughness *σ* = 1.6 μm and load *W* varies from 9 MPa to 39 MPa. As shown in Figure 17a, the oil film thickness decreases from 1.42 μm to 0.6 μm as the load increases. Although the oil film thickness decreases, it can be observed that the trend of oil film thinning becomes slightly flat after the load rises to 24 MPa. This is because the asperities carry a considerable part of the load to stop the oil film thinning trend. As shown in Figure 17b, when the load is small, the oil film is thicker and oil mainly carries the load; as the load increases, the percentage of load carried by asperities increases from 14% to 21% and the area of asperity contacts increases from 2% to 7.5%. The variation of friction and its components with load is shown in Figure 17c. When the ultimate load is not reached (the boundary film does not break), the friction is mainly provided by the fluid and boundary film and the dry friction component is almost 0. When the load exceeds the ultimate load (the boundary film breaks), the total friction increases from 3000 N to about 7500 N, of which the dry friction is 6200 N, about 82.7% of the total friction. Combined with Figure 17d, when the load fails to break the boundary film, the friction pair can maintain a good lubrication state and the friction coefficient remains below 0.02. When the load rises to 34MPa, the boundary film breaks rapidly, the breakdown rate increases to nearly 48% and the friction coefficient rises from 0.02 to 0.06.

From the above analysis, it can be concluded that the excessive load causes the boundary film breakdown and a sudden increase in the friction coefficient. Therefore, selecting a suitable load range is essential for ensuring that the friction pair is under a good lubrication state.

### 5.5. The Influence of Temperature

It is necessary to explore the effect of temperature because it affects both the oil viscosity and the boundary film strength. Figure 18 shows the effect of temperature on lubrication performance with the following working conditions: speed *N* = 100 r/min, load *W* = 36 MPa, roughness *σ* = 1.13 μm and temperature *T* increases from 20 °C to 110 °C. From Figure 18a, it can be seen that the temperature greatly influences the oil film thickness. The minimum oil film thickness *h_min_* is about 1.12 μm when the temperature is 20 °C. When the temperature rises to 110 °C, the oil film decreases to 0.21 μm, and the film thickness ratio also shows the same trend. This phenomenon occurs because as the temperature rises, the viscosity of the lubricant decreases, which enhances the fluidity of the oil. As a result, the dynamic pressure effect is weakened, causing the oil film to become thinner. At the same time, the percentage of load carried by asperities rises from 6.8% to about 38% and the contact area rises from 2.3% to about 14%, as shown in Figure 18b. Figure 18c depicts the variation of friction and its components with temperature. It is clear that when the temperature is below 60 °C, the friction is at a low level of only 2865 N, which is mostly given by the fluid and boundary film. This phenomenon occurs because, when the temperature is low, the lubricant viscosity the higher and it is easier to build an efficient oil film. Moreover, lower temperatures can cause the boundary film to form a stable condition that is difficult to break, with nearly no dry friction. When the temperature reaches 60 °C, the boundary film begins to break, which is accompanied by dry friction. The friction increases rapidly with the dry friction component dominating. When the temperature reaches 110 °C, the friction force climbs to 9775 N, with the dry friction force accounting for 6291 N, about 64.4% of the total friction force. As shown in Figure 18d, when the temperature is lower than 60 °C, the friction coefficient is not more than 0.025 and the breakdown of boundary film almost does not occur. When the temperature rises to 110 °C, the friction coefficient rises to 0.11 and the breakdown rate of the boundary film reaches 27.5%.

As a result, heat dissipation should be ensured during friction pair operation. Excessive temperature will reduce both the bearing capacity of the oil and the strength of the boundary film, resulting in a poor lubrication state for the friction pair.

## 6. Conclusions

Some conclusions drawn from this study are as follows:The test results show that film thickness, pressure, shear stress and temperature are the key factors affecting the strength of boundary film. This paper developed an empirical boundary film strength model, and its parameters can be obtained from the test results, which is easier for engineering applications.A mixed lubrication model considering the boundary film strength was established. It can predict the transition of lubrication status and effectively reflect the tribological characteristics and action ratios of the fluid, boundary film and dry friction components under different working conditions.Low speed and low viscosity are not conducive to the formation of oil film. Properly increasing speed and viscosity can avoid the breakdown of boundary film and metal-to-metal direct contact. When the speed is above 200 r/min or the viscosity is higher than 0.09 Pa·s, the boundary film breakdown rate is almost 0 and the friction coefficient is lower than 0.02.The roughness of the contact surface plays an essential role in mixed lubrication. Reducing the contact surface roughness can improve the lubrication performance and ultimate load of the friction pair. When the roughness is reduced from 1.8 μm to 0.8 μm, the ultimate load of the journal bearing rises from 27 MPa to 36 MPa, an increase of about 33%.When the load exceeds 36 MPa or the temperature is higher than 100 °C, more than 25% of the boundary film breaks and the dry friction component accounts for more than 60% of the total friction, which leads to a sudden increase in the friction coefficient. Therefore, it is important to select an appropriate load range and ensure sufficient heat dissipation for the friction pair.

## Figures and Tables

**Figure 1 materials-16-01035-f001:**
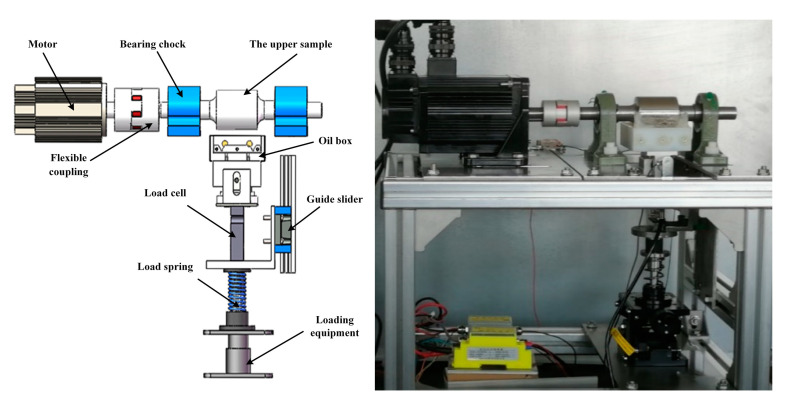
Schematic diagram of double-point contact friction test machine.

**Figure 2 materials-16-01035-f002:**
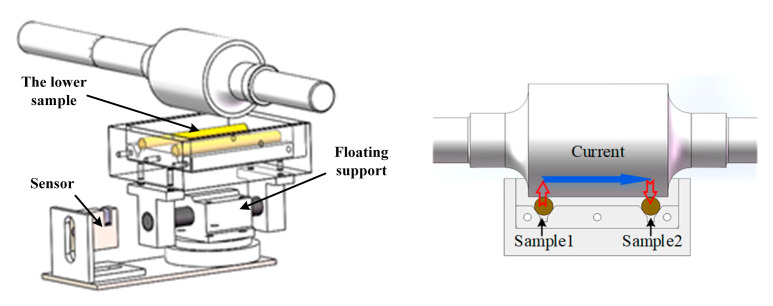
Schematic diagram of sample installation.

**Figure 3 materials-16-01035-f003:**
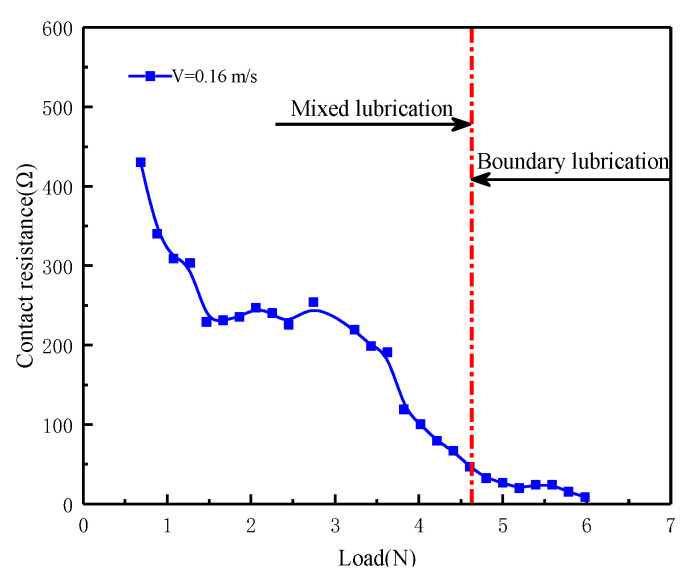
Variation of contact resistance with load.

**Figure 4 materials-16-01035-f004:**
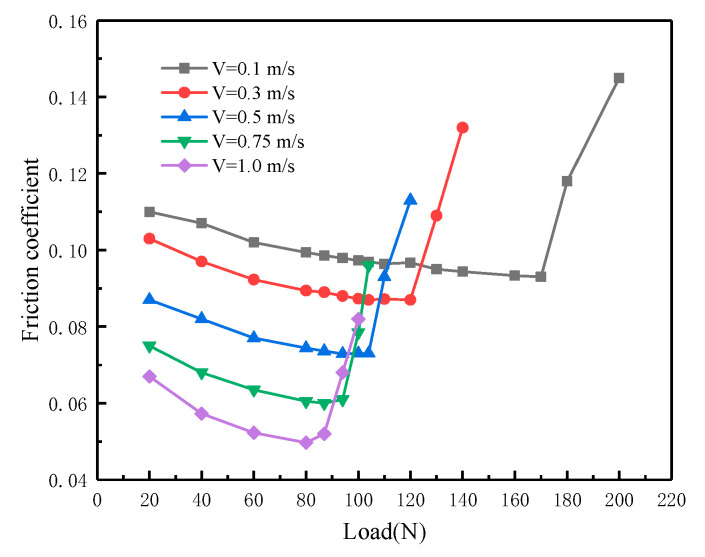
Variation of friction coefficient with load.

**Figure 5 materials-16-01035-f005:**
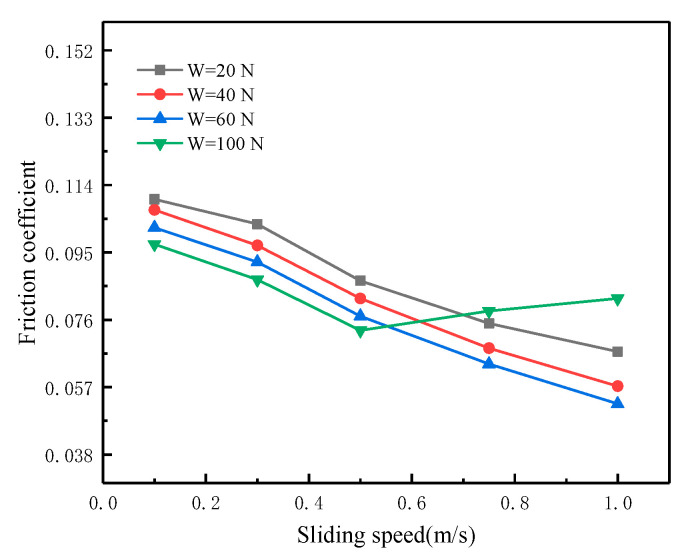
Variation of friction coefficient with sliding speed.

**Figure 6 materials-16-01035-f006:**
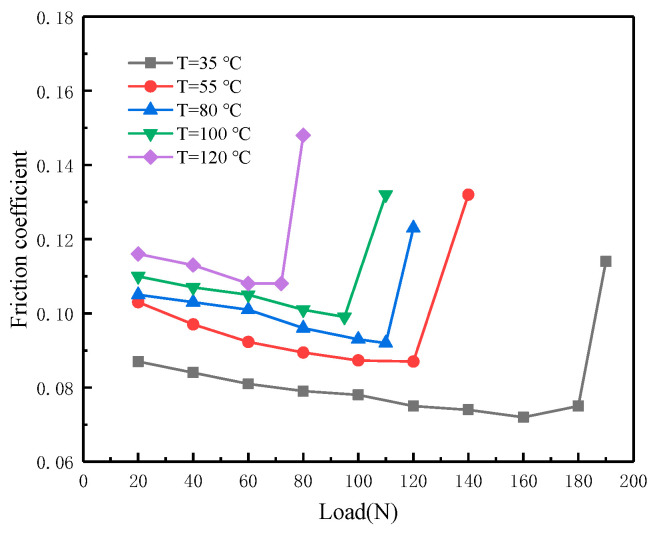
Variation of friction coefficient with temperature.

**Figure 7 materials-16-01035-f007:**
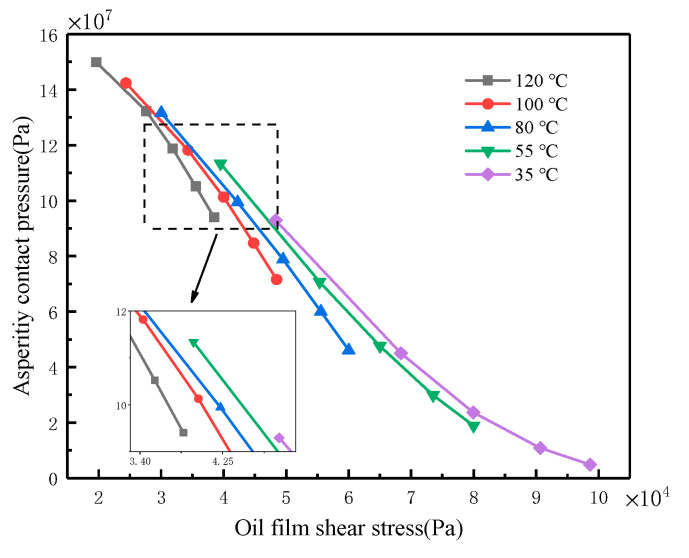
Relationship between asperity contact pressure and shear stress at different temperatures.

**Figure 8 materials-16-01035-f008:**
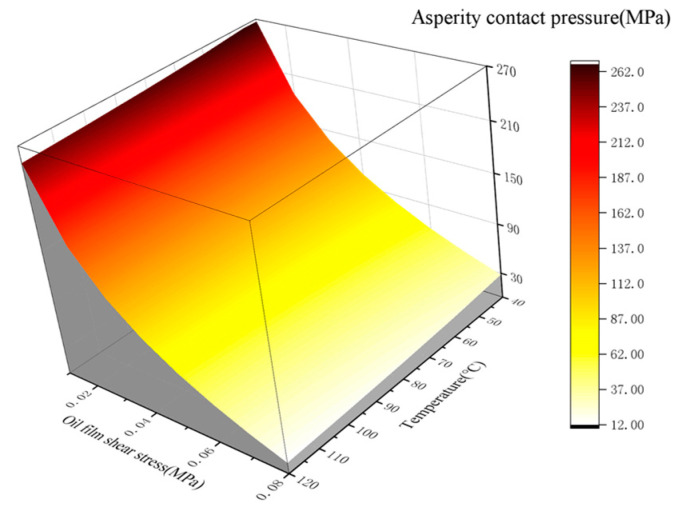
Model curved surface of boundary film strength.

**Figure 9 materials-16-01035-f009:**
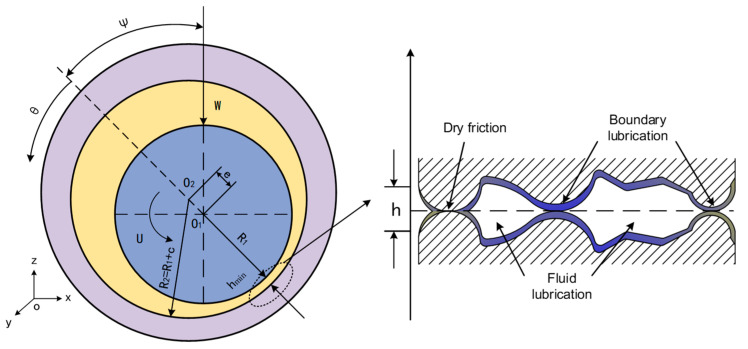
Diagram of journal bearing.

**Figure 10 materials-16-01035-f010:**
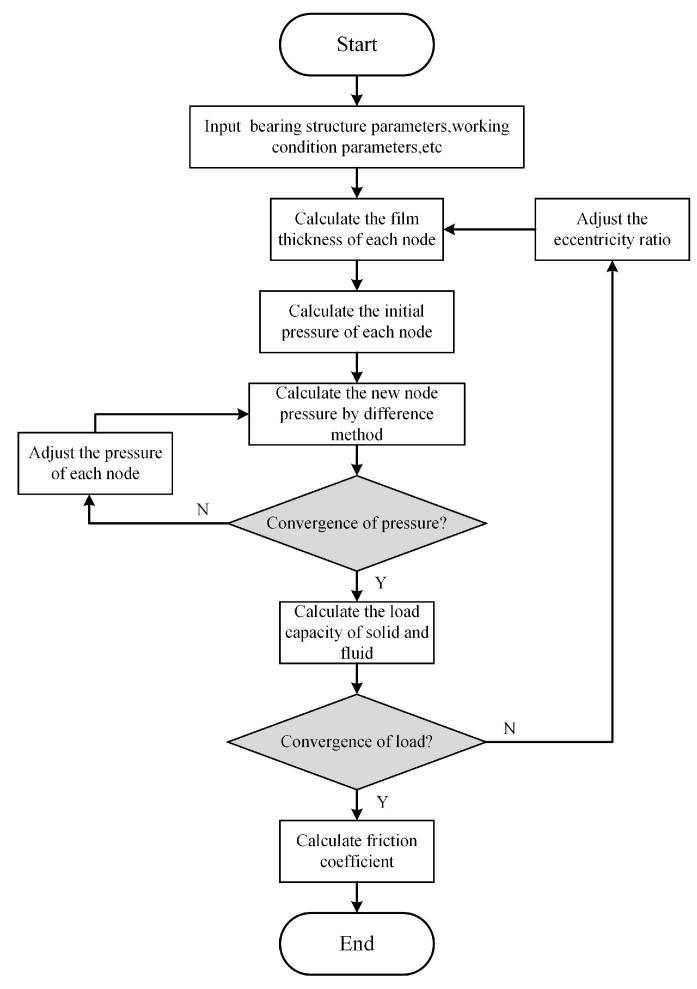
Flowchart of numerical solution.

**Figure 11 materials-16-01035-f011:**
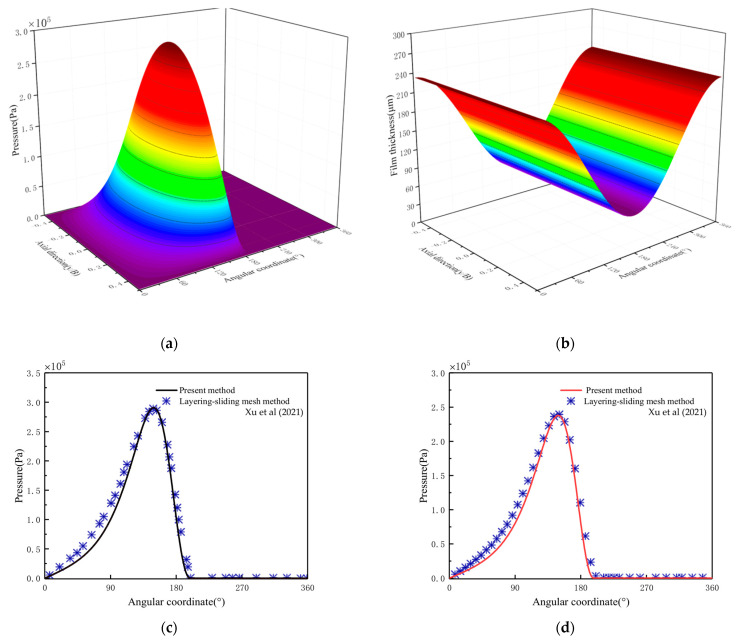
Comparison of calculation results of journal bearing. (**a**) Pressure distribution; (**b**) Film thickness distribution; (**c**) Central pressure; (**d**) Pressure at the location of b/4 [39].

**Figure 12 materials-16-01035-f012:**
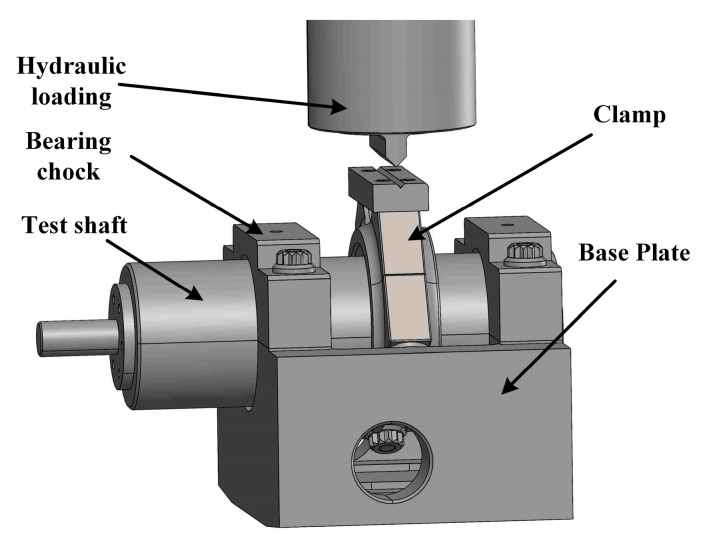
Schematic diagram of the loading structure.

**Figure 13 materials-16-01035-f013:**
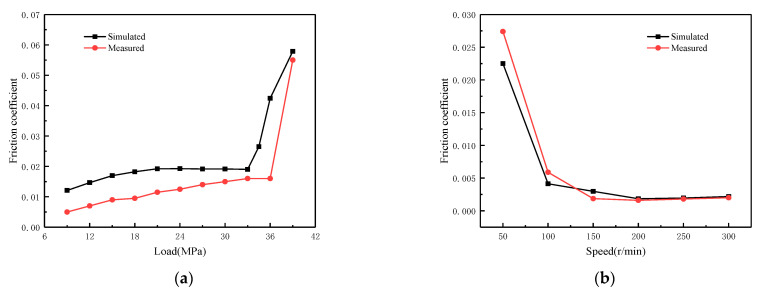
Comparison of theory and experiments. (**a**) T = 55 °C; (**b**) T = 70 °C.

**Figure 14 materials-16-01035-f014:**
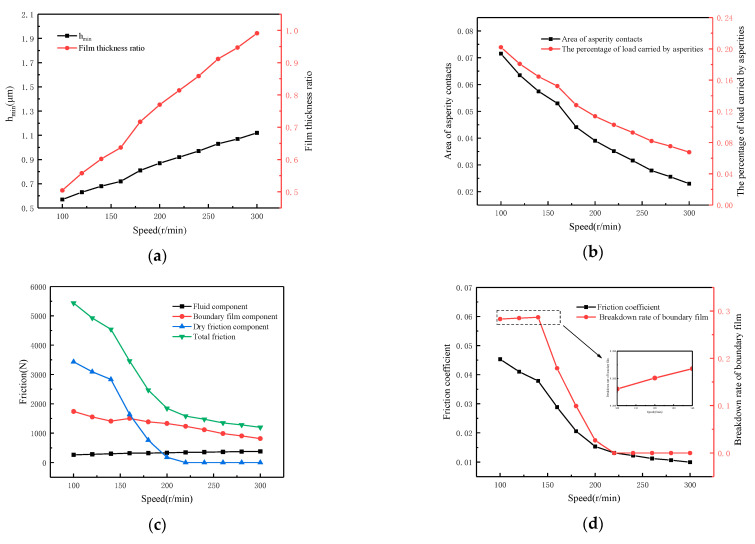
Influence of sliding speed on lubrication performance. (**a**) Calculated film thickness; (**b**) Calculated asperity contacts; (**c**) Calculated friction components; (**d**) Calculated friction coefficient.

**Figure 15 materials-16-01035-f015:**
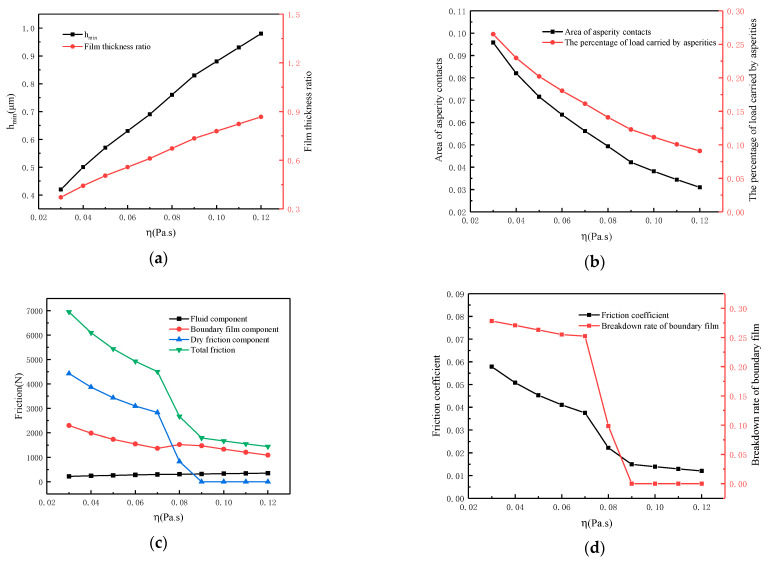
Influence of viscosity on lubrication performance. (**a**) Calculated film thickness; (**b**) Calculated asperity contacts; (**c**) Calculated friction components; (**d**) Calculated friction coefficient.

**Figure 16 materials-16-01035-f016:**
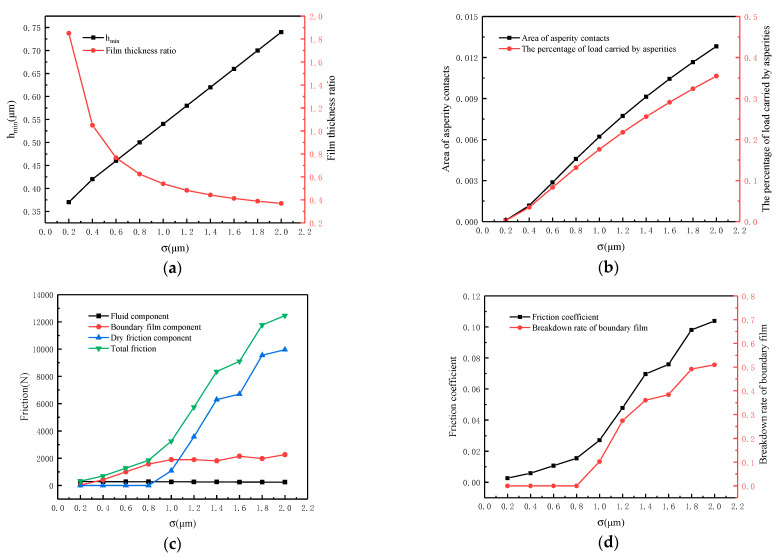
Influence of surface roughness on lubrication performance. (**a**) Calculated film thickness; (**b**) Calculated asperity contacts; (**c**) Calculated friction components; (**d**) Calculated friction coefficient; (**e**) Calculated ultimate load.

**Figure 17 materials-16-01035-f017:**
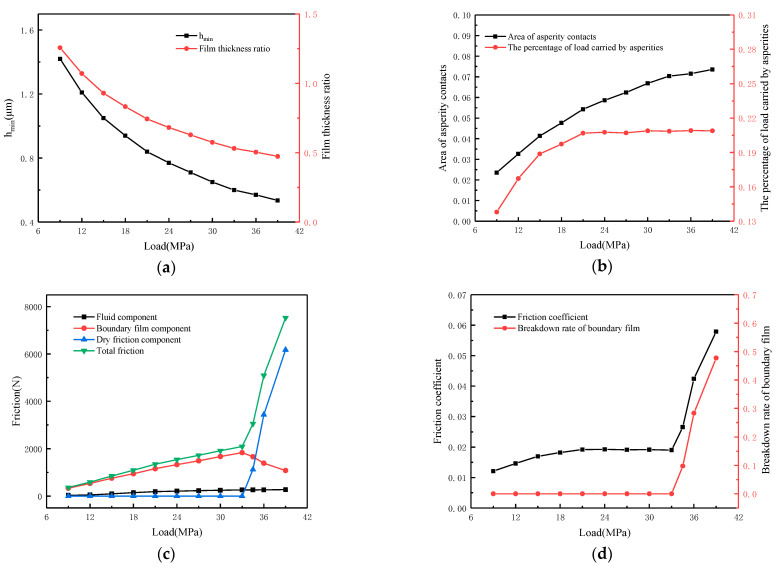
Influence of load on lubrication performance. (**a**) Calculated film thickness; (**b**) Calculated asperity contacts; (**c**) Calculated friction components; (**d**) Calculated friction coefficient.

**Figure 18 materials-16-01035-f018:**
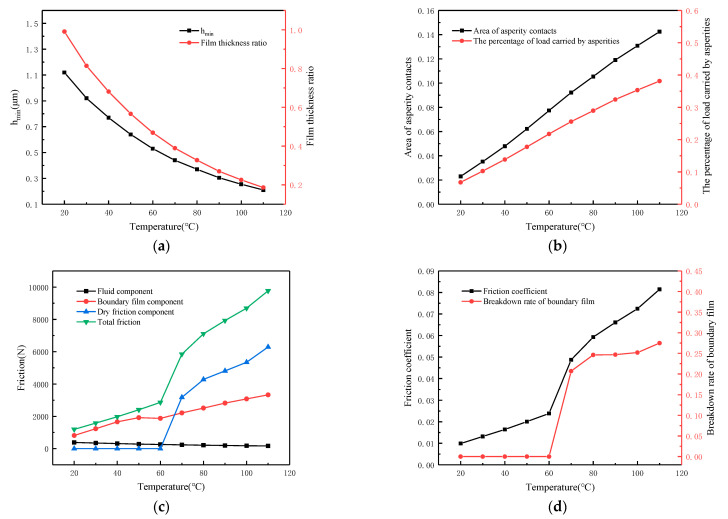
Influence of temperature on lubrication performance. (**a**) Calculated film thickness; (**b**) Calculated asperity contacts; (**c**) Calculated friction components; (**d**) Calculated friction coefficient.

**Table 1 materials-16-01035-t001:** Parameters of the test.

Parameter	Value
Radius of the upper sample, *R_x_*	0.025 m
Radius of the lower sample, *R_y_*	0.004 m
Equivalent elastic modulus, *E*	148 GPa
Surface RMS roughness, *σ*	0.59 μm

**Table 2 materials-16-01035-t002:** Simulation parameters of journal bearing.

Parameter	Value
Bearing radius, R	0.05 m
Bearing width, b	0.133 m
Speed, N	459 r/min
Radius clearance, c	0.000145 m
Lubricant viscosity, *η*_0_	0.0127 Pa·s

**Table 3 materials-16-01035-t003:** Experimental parameters.

Parameter	Value
Load, *W*	9~38 MPa
Speed, *N*	50~300 r/min
Bearing radius, *R*	0.05 m
Bearing width, *b*	0.033 m
Equivalent elastic modulus, *E*	105 GPa
Surface RMS roughness, *σ*	1.13 μm
Radius clearance, *c*	0.0001 m
Ambient temperature, *T*_0_	55 °C, 70 °C
Lubricant viscosity, *η*_0_	0.05, 0.033 Pa·s
The friction coefficient of the boundary film, *μ_a_*	0.1
Eyring stress of the lubricant, *τ*_0_	10 MPa

## Data Availability

The data that support the findings of this study are available from the corresponding author upon reasonable request.

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
