# Peer review of "Experimental and Numerical Study of the Mixed Lubrication Considering Boundary Film Strength"

_materials, 2023, doi:10.3390/ma16031035_

Round 1

Reviewer 1 Report

This is an interesting and timely paper. Improving the understanding of mixed/boundary friction and accurately predicting friction in the mixed/boundary regime key to improving the energy efficiency of machines. 

I have a few comments that the authors may wish to consider to improve their paper still further. 

1. In section 2.3 (Test Specifications) the authors do not seem to use a "running-in" procedure. Such a procedure is commonly used by other researchers to ensure the anti-wear film is adequately developed before friction testing. The authors should state whether a "running-in" procedure was used or not. In addition, in section 2.2, the authors state that a CF-4 SAE 10W-40 lubricant was used. Presumably this means that the anti-wear film considered in the paper is ZDDP. 

2. The Greenwood and Tripp model has been used to estimate mixed/boundary friction. Recently, a number of papers have reported that the Greenwood and Tripp model underestimates the amount of mixed/boundary friction in real contacts (R.I. Taylor et al, Tribology International, 2022, https://doi.org/10.1016/j.triboint.2022.107836 and also M. Leighton et al, Meccanica, 2017, https://doi.org/10.1007/s11012-016-0397-z) and experimental data seems to back this conclusion up (J. Dawczyk et al, Tribology Letters, 2019, https://doi.org/10.1007/s11249-019-1148-9). The authors should mention and discuss the implications of this recent work and how it would affect their conclusions. 

3. In Section 4.2, equation (6) is used to predict the friction coefficient versus lambda. I think a value of 0.5 for the friction coefficient when the boundary film breaks is far too high. There are many published papers on the friction of lubricants that do not contain boundary films, and even when the lambda ratio is very low, the coefficient of friction is usually in the range 0.1 to 0.2. The authors should justify why a friction coefficient of 0.5 is applied. 

4. The model is applied to a journal bearing - this is probably the one component in an engine that is least likely to encounter mixed/boundary lubrication. The authors would, in my view, be better at applying the model to the valve train or the piston assembly. In a journal bearing, at high loads, there will be bearing deformation, and in addition, the pressure increase will increase the viscosity of the oil in the contact, and both effects will tend to increase the oil film thickness compared to that predicted for a rigid bearing. These effects are not taken into account by the authors for their journal bearing model. 

Author Response

Thank you for your letter and for the reviewer's comments. Please see the attachment.

Reviewer 2 Report

The work carried out by the authors is interesting of to the research community but it is required to incorporate the suggestions/comments in their manuscript to improve further for better understanding.

1.     In the abstract, the author should explain about the experiment and their finding then correlate with numerical.

2.     The author must look the English in terms of punctuation, grammar, etc.

3.     The author can look into the manuscript for lubrication mechanism and shear stress model in mixed regime https://doi.org/10.1007/s11249-017-0863-3

4.     What are the calibration standards of the designed setup?

5.     What is the surface roughness of the counter body before the test?

6.     In conclusion author should include experimental results also.

Author Response

(The authors gave the same response as above.)

Reviewer 3 Report

Title: Mixed Lubrication of Sliding Friction Pairs Considering Boundary Film Strength: Experimental Study and Numerical Modeling

The following comments have been made to improve the quality of the manuscript.

1)    The title needs revision. It should be in such a way that it should depict the exact work. A good title contains the fewest possible words that adequately describe the contents of your research work.

2)    While reading the abstract, I didn’t see the exact objective of the work. Please be concise with the objective.

3)    Further, a sufficient description of the simulation methodology and outcome should be in the abstract.

4)    Please mention the specific outcome (such as in percentage, number, temperature, etc) at the end of the abstract.

5)    The literature review is written well but the research gap is not mentioned clearly.

6)    The last passage of the introduction needs rephrasing. Please check lines 78-84.

7)    Where are CFD governing equations and boundary conditions etc?

8)    Where are geometric details/dimensions in the figure?

9)    What type of mathematical model was used?

10)        Did you study the grid independence test?

11)        Did you validate the numerical model?

12)        Did you use user-defined functions or any kind of mathematical code?

13)        The conclusion is insufficient. You should highlight a few particular results of this work.

14)        There are many grammatical mistakes in this work.

Author Response

(The authors gave the same response as above.)

Round 2

Reviewer 1 Report

The manuscript has been improved substantially and suggestions made in the earlier review have been incorporated. I still believe it would be better, in equation (8) to define the friction coefficient at high pressure as being fo and state that the value of fo used in the paper is 0.5. This would enable the model to be used by other researchers that did not wish to use the dry friction contact value. 

Author Response

(The authors gave the same response as above.)

Reviewer 3 Report

Thanks

Author Response

(The authors gave the same response as above.)
